# Calcined Clays from Nigeria—Properties and Performance of Supplementary Cementitious Materials Suitable for Producing Level 1 Concrete

**DOI:** 10.3390/ma16072684

**Published:** 2023-03-28

**Authors:** Abubakar Muhammad, Karl-Christian Thienel, Sebastian Scherb

**Affiliations:** Institut für Werkstoffe des Bauwesens, Universität der Bundeswehr München, 85577 Neubiberg, Germany

**Keywords:** clay mineralogy, calcined clay, hydration mechanism, ion solubility, workability, strength activity index, metakaolin, metasmectite

## Abstract

In this work, four naturally occurring (two kaolinite-rich and two smectite-rich) clay samples were collected from different areas around the Ashaka cement production plant, located in Gombe State, Nigeria and calcined in a laboratory. The mineralogical characterization of the clays was carried out by XRD. The hydration kinetics of the calcined clay–cement systems were monitored by isothermal calorimetry. Workability was determined using the flow table method. The reactivity of the calcined clays was determined from the solubility of Si and Al ions and the strength activity index. All calcined clays studied met the requirements of ASTM C618 for the use of natural pozzolans as a partial replacement for hydraulic cement. The metasmectite clays yielded a higher specific surface area, increased water demand, and less reactive Si and Al ions compared to the metakaolin clays. The two calcined clay groups require the addition of superplasticizer to achieve a workability class similar to the Portland cement mortar system. They can be used to replace Portland cement at replacement levels of up to 45%, in combination with limestone powder to form an LC^3^ cement, thereby achieving at least a “Level 1” reduction in greenhouse gas emissions.

## 1. Introduction

Demand for shelters and other civil engineering infrastructure has recently increased in Nigeria and other African countries due to rapid urbanization and a high annual population growth rate [1,2]. To meet the country’s infrastructural requirements, the Federal Government of Nigeria in partnership with private investors embarked on massive infrastructural development, such as construction of the Abuja world trade center, Eko Atlantic, and Lekki free trade zone. These projects and others ongoing in the country consume a large quantity of cement annually. The production of cement, an essential component of concrete, presents two major challenges. First, a large amount of energy is needed for its production and grinding. This is a problem especially in developing countries, making cement more expensive. Second, cement production accounts for about 6% to 10% of the global carbon emissions [2]. Although the total per capita carbon emissions of sub-Saharan Africa in 2018 were only 0.81 kt, below the 2.7 kt threshold for limiting global warming to below 2 °C by 2050 [3], the rate of urbanization in the region is increasing rapidly compared to the rest of the world, leading to an increase in demand for shelter and other concrete infrastructure, with a consequently higher demand for cement [1]. For example, the cement production capacity of Nigeria—the most populous country in the sub-Saharan region—was only 21 Mt in 2015, but increased to 28 Mt by 2020, and it is expected to reach 53 Mt in 2040 [4]. Although total carbon emissions from cement production were estimated at 11 MtCO_2_e in 2015, they increased to 15 MtCO_2_e in 2020 and are expected to reach 28 MtCO_2_e in 2040 [4].

In order to continue to benefit from the use of concrete as the main construction material and to limit its environmental impact, the use of concrete with lower embodied carbon is necessary. One of the potential steps in the route map to achieve this is reducing the amount of clinker used per m^3^ of concrete, as this is the main factor of embodied carbon in concrete [5]. Therefore, partial replacement of cement with supplementary cementitious materials (SCMs) that require less energy for production and release less CO_2_ is essential to help limit global warming [6]. One of these SCMs that has the potential to replace a high percentage of cement is calcined clay (CC) [2]. This material is particularly suitable for Nigeria with a lower supply of conventional SCMs such as fly ash, slag, and silica fume [7].

Common clays can be used as reactive SCMs after thermal activation [8,9,10]. The mineralogical diversity consisting of activatable phyllosilicates (e.g., kaolinite, illite, and smectite) and accompanying minerals (e.g., quartz and feldspars) inevitably requires an individual characterization (e.g., mineralogical and chemical composition) of each clay deposit to verify its suitability as SCM [11]. The R^3^ test [12,13] or the determination of the solubility of silicon and aluminum in alkaline solution [14,15] has been shown to be reliable for assessing the reactivity of CC [11,16]. As an indirect method, determining the strength activity index (SAI) has been shown to be accurate for evaluating the pozzolanic potential of CC as a partial replacement for cement [9]. In addition to mineralogical and chemical composition and the reactivity of the CC, physical properties such as water demand are important factors that significantly influence the possibilities of high replacement levels [17]. The higher water demand and higher specific surface area of CC in comparison to cement negatively affect the workability of the system [18], which is an important attribute, especially when considering the ease of compactability, which in turn affects strength and durability of mortar and concrete [19].

The pozzolanic potential of a CC for its intended use in concrete and mortar can be evaluated according to the provisions of ASTM C618 [20]. To obtain the relative performance of the CC blended cement, 20 wt.% ordinary Portland cement (OPC) is substituted with CC, and the w/p is adjusted to achieve a flow diameter similar to OPC system. In this study, a constant volumetric proportioning system was used as previously described in [21]. OPC was replaced on a volumetric basis while maintaining the same amounts of limestone powder (LP), sand, and water. SP was added to the CC mortar system to achieve a workability class similar to that of the OPC mortar system.

The great potential of CC is based on physical effects such as the filler effect [22,23] and the adsorption of ions from the pore solution [24], as well as on the release of silicon and aluminum ions at an early time of hydration [15,25]. This is evidenced by a strong influence of the early clinker hydration in CC blended systems [26,27,28]. The released aluminum exploits its full potential, especially in the presence of LP [18,29,30]. When CC and LP are blended and used as a substitute for clinker, they promote the formation of monocarboaluminate/hemicarboaluminate AFm phases and densify the concrete microstructure, which in turn have positive effects on the mechanical and durability properties of mortar and concrete [18,31,32,33].

Using the technology of LC^3^ cement [34,35] (50 wt.% ground clinker, 30 wt.% CC, 15 wt.% LP, and 5 wt.% gypsum), which is currently gaining more attention in the global construction market, reactive kaolinite-rich CC (40 wt.% kaolinite and above) achieves a 28 day compressive strength similar to the OPC reference system [36]. However, clays with low kaolinite content should also come into focus due to their economic and ecological advantages and lower competition with other industries. For many simple applications that can solve the housing shortage in the sub-Saharan African region, these clays could be of great interest.

## 2. Research Gap

Numerous investigations have been carried out on the suitability of replacing up to 45 wt.% OPC with a blend of CC and LP forming LC^3^ cement (e.g., [34,36,37]). Some criteria have been established as a basis for the selection of the appropriate CC, which include the mineralogical composition of the raw clay, the oxide composition, its reactivity, and its pozzolanic potentials. There are no previous studies in the literature on the potential use of the CCs presented in this manuscript in LC^3^ cements. Therefore, this work investigates the suitability of four common clays from Gombe State, Nigeria, for use as SCMs to achieve at least a “Level 1” reduction in greenhouse gas emissions. The focus is on determining the mineralogical composition, physical and chemical properties, and effects of CC on the hydration mechanism, workability, and SAI of the OPC.

## 3. Materials and Methods

### 3.1. Selection of the Nigerian Raw Clays 

Nigeria has different cement manufacturers across its regions. Figure 1 shows some of the cement plants, most of which are located near limestone deposits [38], as well as the location of the sampling sites. The potential of clay deposits in the vicinity of some of the cement plants has already been established. For instance, clay from Ogun State has already been studied by [39,40,41]. The current research, therefore, focuses on studying the clay deposits around the Ashaka cement plant. Four raw clay samples were collected from the areas of the Nafada, Kwami, and Billiri local governments of Gombe state and denoted as NRC (Nigerian raw clay)-1 to NRC-4. 

### 3.2. Methods

#### 3.2.1. Calcination and Grinding of NRC

After delivery, the NRCs were dried at 60 °C in the laboratory oven for 24 h and ground in a rotary disc mill (Retsch, RS 200) at 700 rpm for 5 min for further investigation. On the basis of the development of the thermal decomposition presented in Figure 2, the NRCs were calcined for 30 min using a laboratory muffle furnace at a temperature of 750 °C (red lines in Figure 2); after cooling to ambient temperature, they were subsequently ground in a rotary disc mill at 700 rpm for 5 min and labeled after calcination as NCC (Nigerian calcined clay)-1 to NCC-4. From this point on, the term “meta” refers to the calcined material.

#### 3.2.2. Characterization of the Cement, Limestone Powder, and Superplasticizer

The cement used (CEM I 42.5 R) complied with DIN EN 197-1 [43]. It contained (wt.%) 61.6 C_3_S, 18.2 C_2_S, 5.8 C_3_A, 9.0 C_4_AF, 3.2 sulfate, and 0.6 calcite as the mineralogical phases (according to manufacturer’s information). The limestone contained 99.8 wt.% calcite and 0.2 wt.% quartz. SP with a solid content of 38.6 wt.%, anionic charge density of 1390 μmol/g, molecular weight of 25,992 g/mol, and side-chain length of n_EO_ = 31 as specified by the supplier was used. Standard sand according to EN 196-1 [44] was used as fine aggregate for mortar tests.

#### 3.2.3. Characterization of the Raw and Calcined Nigerian Clays

Thermal decomposition of the clay mineral phases was investigated using thermogravimetric (TG) analysis conducted in Netzsch STA 449 F3 Jupiter equipment with a heating rate of 2 °C/min. Further quantification of the mineralogical composition of the NRC was determined by X-ray diffraction using a side-loading preparation in a PANalytical Empyrean diffractometer, with CuKα radiation (1.54 Å) at 40 kV and 40 mA; Rietveld refinement was performed using Profex-BGMN [45]. Since there was no significant amount of amorphous phase in the NRC, the values were normalized to 100%. Inductively coupled plasma optical emission spectrometry (ICP-OES) was used to determine the oxide content of the NCCs after melt fusion as described in [46] in accordance with DIN EN ISO 11885 [47]. The reactivity of the NCCs was evaluated by eluting the NCCs in NaOH solution (10%) for 24 h to determine the solubility of silicon (Si) and aluminum (Al) ions according to the method specified by [48] and described in [46,49].

The determined physical parameters of the NCCs included the particle density measured using a helium pycnometer, according to DIN EN ISO 17892-3 [50], the BET specific surface area measured using a HORIBA SA-9600 series surface area analyzer, according to DIN ISO 9277 [51], particle size distribution (PSD) determined using laser light diffraction (Bettersizer, 3P instrument) [30], and water demand based on the Puntke method [52].

#### 3.2.4. Performance in Mortar and Early Hydration Behavior of the NCC

To evaluate the influence of NCCs on the workability and SAI of blends with OPC, two mortar series were investigated. The first with binary blends containing CC only as a substitute for OPC at 20 vol.%, designated as M—the reference system, with OPC only as a binder, and MN-1 to 4 (mortar produced with NCC-1 to 4 substituting OPC at 20 vol.%). The second batch of the ternary blends, labeled ML, constituted mortars produced with LP substituting OPC at 15 vol.% (later on referred to as Portland limestone cement (PLC)), and MLN-1 to 4 (containing 55 vol.% OPC, 15 vol.% LP, and 30 vol.% of NCC-1 to 4). Table 1 shows the gravimetric mix designs of the different mortar systems. The constituents were mixed in a stainless-steel 5 L bowl mixer according to DIN EN 196-1 [44]. The flow diameter of the fresh mortar was determined according to DIN EN 1015-3 [53]. For the SAI, the mortar was poured into sets of 40 mm × 40 mm × 160 mm steel molds and compacted on a vibrating table according to [44]. The specimens were covered and stored moist for 48 h, and then cured under water. The compressive strength was tested at 7, 28, and 90 days using the Form + Test Prüfsysteme Alpha 1-3000 testing equipment at an increased uniform loading rate of 2400 ± 200 N/s till failure. The SAI for each test age was calculated as the ratio of the compressive strength of the specimens with cement replacement to those without replacement, multiplied by 100 [54].

Early hydration behavior of the OPC, binary blends of OPC and NCC, and ternary blends of OPC, LP, and NCC were monitored through isothermal calorimetric measurements using a TAM AIR eight-channel isothermal calorimeter. Two series of measurements were carried out: the first with 20 wt.% NCC replacing OPC (OPC-20NCC); the second with 55 wt.% OPC, 15 wt.% LP, and 30 wt.% NCC (OPC-15LP-30NCC). Both series were performed with a water-to-binder ratio (w/b) of 0.5. The test procedure is given in [55]. The heat flow was measured up to 48 h, and the result was normalized to 1 g of cement. According to the LC^3^ (CEM II/C-M (Q-LL) [56]) system recipe, the calorimetry was also performed on sulfated systems with an increased content of 5 wt.%.

## 4. Results

### 4.1. Characterization of the Clays

#### 4.1.1. Nigerian Raw Clay Characterization

Figure 2a depicts the differential thermal decomposition of the NRC mineral phases and their accompanied mass loss due to decomposition (Figure 2b). Some similarities and distinctions among the clays could easily be identified. NRC-1 and 2 showed a typical differential thermal analysis (DTG) curve pattern of a smectitic clay, whereas NRC-3 and 4 showed a DTG curve pattern of a kaolinitic clay. All NRCs exhibited mass loss at lower temperatures (100 °C) due to the release of free water. The smectite-rich clays had a greater mass loss between the temperatures of 100 and 300 °C, associated with the removal of water bound to the outer surfaces of the clay minerals. At an intermediate temperature (400 to 650 °C), the kaolinite-rich clays exhibited more dehydroxylation than the smectite-rich clays. On the basis of the results, the calcination temperature for all NRCs was set at 750 °C, between the dehydroxylation and recrystallization temperatures.

Figure 3 displays the diffraction pattern of all four NRCs. The y-axes are shifted in the cases of NRC-1, 2, and 4 (the maximum shift for NRC-1 was 1000 counts). The large smectite peak was clearly visible for NRC-1 and 2. Kaolinite and quartz were visible for all four clays. Quantification of the various mineralogical phases is provided in Table 2. NRC-1 and NRC-2 were smectite-rich clays containing about 70 wt.% smectite and about 13 and 22 wt.% kaolinite, respectively. NRC-3 and NRC-4 were kaolinite-rich clays (56 and 49 wt.%, respectively) and they contained more than 30 wt.% quartz as the second main phase.

Table 3 provides the oxide composition of the four NCCs. The content of sulfur trioxide (SO_3_) and the loss on ignition (LOI) of all clays were within the thresholds specified by ASTM C618 [20]. Furthermore, the summation of the SiO_2_, Al_2_O_3_, and Fe_2_O_3_ contents of all clays was greater than 70 wt.% (the minimum recommended by ASTM C618). Therefore, from the oxide composition point of view, all four NCCs satisfied the minimum requirements specified by [20] for the use of natural pozzolans as cement replacement.

The metakaolin-rich CC had a lower specific surface area than the metasmectite-rich ones. The order of increasing surface area was NCC-3 < NCC-4 < NCC-2 < NCC-1. The variation of the specific surface area had a less significant effect on the water demand and density of the NCC, as shown in Table 4. Similar to the surface area, the same sequence of increasing water demand was observed, with smaller differences between samples. All four NCCs had almost similar density values ranging from 2.66 to 2.85 g/cm^3^. NCC-3 had the finest particle size distribution among the four NCCs with a d_50_ of 13.8 µm, followed by NCC-2 (d_50_ = 21.7 µm) and NCC-4 (d_50_ = 28.2 µm), while NCC-1 (d_50_ = 34.5 µm) had the coarsest particle size distribution.

#### 4.1.2. Reactivity Assessment of the NCC

Figure 4 provides the Si and Al ion solubilities of the NCCs, where the numbers placed above the bars denote the Si/Al ratio. NCC-3 released the most Si and Al ions, followed by NCC-4, NCC-2, and NCC-1. NCC-3 and NCC-4 were metakaolin-rich clays with a Si/Al ratio approximately equal to 1.0. NCC-2 had a Si/Al ratio of 1.49, while NCC-1 had the highest Si/Al ratio of 2.24.

### 4.2. Influence of NCCs and LP on Mortar Proparties

When substituting part of the cement with NCCs, the water demand of the system increased, which affected the workability of the mortar. Figure 5 shows the influence of the CC with and without LP on the flow spread of the mortar. All CC mortar systems demanded an increased amount of water to attain a slump flow similar to OPC and PLC (the reference systems), due to their higher water demand compared to OPC and PLC. Consequently, the w/b ratio must be adjusted or an SP dosage must be added when CC is used as a partial replacement for OPC and PLC, until a slump flow of ±5 cm is achieved compared to the reference mortar mix. In this study, a constant SP dosage of 0.3 and 0.35 wt.% of binder was used to bring both the M and ML systems to a slump flow approximately ±5 cm of the reference mortar. In both systems, the effect of the NCC on the flow diameter of the mortar was in the following order: NCC-1 > NCC-2 > NCC-4 > NCC-3. However, this effect across the NCCs is rather narrow, as there were no significant differences in slump flow values. Therefore, both the metakaolin and the metasmectite systems had a similar effect on the mortar workability.

The pozzolanic activity of all four NCCs at 20 vol.% cement replacement was quite similar at 7 days and exceeded the minimum of 75% SAI of the OPC as recommended by ASTM C618 [20]. After 28 days of curing, the SAI improved with a clear distinction across the four NCCs, which could be attributed to the differences in the solubility of the Si and Al ions, as depicted in Figure 4. At 28 and 90 days, the MN-3 mixture attained a SAI of more than 100% as a binary blended system. This is due to the higher solubility of Si and Al ions in NCC-3 compared to other NCCs (Figure 3). With the ternary blend of 30 vol.% NCC and 15 vol.% LP in the MLN mortars, lower compressive strength values were achieved, and the SAI fell below the recommended 75% SAI value, as shown in Figure 6b. After 28 and 90 days of curing, only the metakaolin-rich NCC mortar systems achieved the minimum SAI of 75%, while the SAI of the metasmectite-rich NCCs fell below the recommended value of 75%. 

The actual performance of the NCC blended cement mixes compared to PLC is shown in Figure 7. Here, all NCC blended cements achieved a pozzolanic activity index >75% after 28 days of curing, making them suitable as substitutes for PLC in concrete and mortar. Again, the mortar made with NCC-3 exceeded the reference at 28 and 90 days resulting, in a SAI greater than 100%.

### 4.3. Influence of NCC on Early Hydration

Heat flow curves obtained for OPC-20NCC and OPC-15LP-30NCC are presented in Figure 8a and b, respectively. While the addition of 20 wt.% NCC led to a comparable acceleration of the silicate reaction for all four NCCs (occurrence after about 9 h), a significant time shift could be observed for the aluminate reaction. Comparing the physical parameters of the NCCs, no correlation could be drawn between the PSD or BET surface area and the acceleration of the silicate and aluminate reaction. Despite considerable differences, the physical filler effect seemed to be comparable for all NCCs. Maier et al. [24] assumed the agglomeration effects of CC particles as one of the reasons for the different behavior of CC compared to limestone powders. Such agglomerations could also be a reason for the small differences in silicate reaction among the NCCs. The formation of the aluminate peak occurred after 25 h (NCC-1) and 20 h (NCC-2) in the mixtures with metasmectite-rich CC and after 16 h (NCC-3) and 14 h (NCC-4) in the mixtures with metakaolin-rich CC. Overall, a clear distinction could be made between silicate and aluminate reactions, and the onset of the aluminate reaction [57] was clearly visible for all 20NCC systems. 

The addition of 15LP30NCC led to a pairwise similar curve shape of the metasmectite- and metakaolin-rich samples (Figure 8b). While the end of the dormant period was comparable for all four NCCs, the aluminate reaction was strongly accelerated to such an extent that it overlapped with the silicate reaction between 8 and 9 h, and no onset of the aluminate reaction was detectable for the metakaolin-rich NCCs, indicating an undersulfated system, as previously observed by [23,24,28]. For the metasmectite-rich CC, the differentiation between the silicate reaction (between 8–9 h) and the aluminate reaction after 14 h was still possible.

The current study considered, in addition to the adsorption of SO_3_ on CSH phases [22,57], the adsorption of ions from the pore solution onto the negatively charged surface of the CC particles [58] as a major reason for the different shift in aluminate reaction [24,59]. In this context, various metaclay minerals exhibit markedly different surface properties (e.g., zeta potential), as shown by Schmid and Plank through interactions with superplasticizers [58].

Figure 9 displays the influence of further substituting the OPC with 5 wt.% G (G = gypsum) on the hydration behavior of OPC-15LP-30NCC systems. The authors are aware that 5 wt.% gypsum is the maximum allowable amount. Nevertheless, an additional 5 wt.% was added to the 1.6 wt.% already present in the OPC to monitor the degree of sulfation required by different OPC-15LP-30NCC binder systems. The additional gypsum led to a shift in the aluminate reaction toward 23–24 h for the metakaolinite-rich systems. This was very well balanced compared to OPC-15LP. In comparison, no clear aluminate peak could be identified in the metasmectite-rich samples due to the further addition of gypsum, already indicating a too high sulfate dosage for these systems. The lower sulfate requirement for 2:1 compared to 1:1 meta-phyllosilicates is in line with previous studies [26].

The heat of hydration values per gram of cement after 48 h (Table 5) revealed a significant increase with the addition of both 20NCC and 15LP-30NCC compared to the respective reference mixture. Within the series of measurements, however, the different NCCs exhibited hardly any differences. Clear differences could only be detected between the metakaolin- and smectite-rich systems when gypsum was added (OPC-15LP-30NCC-5G). The metakaolin-rich samples had about 20% higher heat of hydration compared to the metasmectite-rich samples. This could have been due to the strong overlapping of the silicate and aluminate reaction in the undersulfated systems and, thus, a higher reaction rate of the clinker phases for the balanced system. Since metakaolin is able to independently form a high content of hydrate phases at early times of hydration [15], the higher heat of hydration could also be explained by a chemical contribution of metakaolin and, thus, additional hydrate phase formation. Hence, the additional gypsum provided an improvement in performance only for the metakaolin-rich samples, while no improvement could be derived for metasmectite-rich samples in terms of the heat of hydration.

## 5. Discussion

Among the four clays studied in detail, two were kaolinite-rich, while the other two were smectite-rich clays. Previous studies [29,36] revealed the possibility of substituting up to 45 wt.% cement with a blend of 15 wt.% LP and 30 wt.% reactive metakaolin. Reactive metakaolin is considered particularly suitable in this context due to its early age reactivity potential. The two metakaolin-rich CCs (NCC-3 and NCC-4) released the most reactive silicon and aluminum ions compared to the metasmectite ones, obviously indicating good pozzolanic potential [60,61]. Both groups of CCs could be used as partial replacement of cement in order to obtain decarbonized concrete.

Partial replacement of cement with CC is one of the potential pathways to achieve a clinker-efficient cement [4]. Although clinker is the most important component of cement and contributes greatly to the strength formation of concrete, its production is the main driver of the embodied carbon in concrete [62]. The current challenge is how to further decarbonize concrete without compromising its engineering properties such as workability, strength, and durability. In this context, the performance of NCC mortar compared to OPC can be explained in two aspects. 

First, the actual performance of the NCCs in relation to the compressive strength development and their effect on the workability of the mortar qualify them for use as SCMs in concrete. Both the binary and the ternary blends of NCCs and LP reduced the workability of the mortar, requiring the addition of up to 0.35% SP to achieve similar performance to the OPC and PLC mortar reference system (Figure 5). This would incur additional but negligible costs considering the ≥20 vol.% and replacement of cement with a less costly material. Moreover, considering development of the compressive strength, the binary mix with CC (20 vol.% cement replacement) had less impact on the compressive strength development after 28 days of curing. The minimum strength achieved was 44 MPa for the OPC-NCC composite cement, which is enough for use in plastering and general production of concrete up to grade C40/50, except for the production of precast elements which require a higher early strength [63,64] than achievable with these blends without further measures [65]. For the ternary composite cements, the same application holds for the metakaolin-rich NCCs, whereas for the metasmectite-rich NCCs, the compressive strength after 28 days of curing was 36 and 38 MPa for MLN-1 and 2, respectively. This limits their use to general construction work under normal environmental conditions and the production of concrete up to grade C35/45. In practice, this concrete grade is sufficient for general construction works to provide the required concrete infrastructure in the sub-Saharan African region. Thus, it can significantly reduce the cost of cement and increase the affordability of houses.

Second, the performance of CC and LP can be used as a lever for the decarbonization of cement, which can help to reduce the cement industry’s contribution to CO_2_ emissions. The cement industry contributes about 6–7% of global CO_2_ emissions, of which two-thirds are process-related (CO_2_ released during calcination of limestone) and one-third are fuel-related [62]. The average greenhouse gas emissions from the use of CEM I for the production of C25/30 grade concrete are estimated at 237 kg CO_2_ equivalent/m^3^ of concrete [66]. Substituting this cement with CC at 20 vol.%, which releases only about 40% of the CO_2_ released by OPC [67,68,69], results in a CO_2_ reduction of 12%. For a ternary blend of 30% cement with 15% LP, a CO_2_ reduction of about 30% is achieved. Therefore, from an environmental view point, the ternary blend of NCC, LP, and OPC achieved at least a “Level 1” reduction in the greenhouse gas emissions, according to the concrete sustainability council classification [66]. This makes it a suitable material for general concreting works in the sub-Saharan African region, where a high volume of construction work is expected in the near future. 

As previously reported [18,70], the use of CC as a partial replacement for cement favors the precipitation of hemicarboaluminate phases from the aluminate and ettringite reaction, which occurred earlier and more broadly in the CC studied here in the following order: NCC-4 > NCC-3 > NCC-2 > NCC-1. A clear distinction also existed between the silicate and aluminate hydration maxima in all CC blended systems, except NCC-4, which had a strong overlap of these maxima. Therefore, at 20 wt.% CC replacement, only NCC-4 requires a sulfate adjustment to prevent flash setting and ensure sequential occurrence of the hydration maxima. For the ternary mixtures, the hydration kinetics of the blended cement changed. Here, the silicate and the aluminate peaks were merged together due to the rapid dissolution of sulfate in all ternary binders and occurred earlier in the metakaolin systems than in the metasmectite systems [24]. Similar to NCC-4 binary binder systems, proper sulfation is urgently required in the ternary blend CC systems.

## 6. Conclusions

The contents of SO_3_ and LOI, as well as the summation of the SiO_2_, Al_2_O_3_, and Fe_2_O_3_ contents in all clays, were within the limits recommended by ASTM C618 for natural pozzolan to be used in concrete.

Although the metakaolin-rich CC released more reactive Si and Al ions compared to the metasmectite-rich CC, the metasmectite-rich CC also yielded relatively good reactivity both in a 20 vol.% binary mixture with cement and in a ternary mixture in the presence of LP with a combined 45% OPC substitution, as measured by isothermal calorimetry.

The metasmectite-rich CC had a larger specific surface and a higher water demand than metakaolin-rich CC. This would affect the workability of the mortars and necessitate the use of SP to achieve the same flow diameter as the OPC system.

In binary binders using 20 vol.% cement replacement, all CC mortar systems achieved a SAI greater than 75% at 7 days of hydration onward, the minimum value recommended by ASTM C618 for a natural pozzolan to be used as a pozzolanic cement replacement. 

For the ternary mix containing 15 vol.% LP and 30 vol.% CC, in addition to OPC, the metasmectite-rich CC mortar systems did not reach the minimum SAI of 75% within 90 days when compared with OPC. The metakaolin-rich CC mortar systems, on the other hand, achieved the minimum requirement after only 28 days of curing.

Sulfate adjustment was not required for the binary CC binder systems except for NCC-4 and all ternary blended CC and LP systems, which require urgent sulfate adjustment to avoid flash setting and maintain the sequential formation of hydration peaks. 

The Nigerian calcined clays NCC-1, NCC-2 and NCC-3 can be used as direct replacements for 20 vol.% of OPC without sulfate adjustment. For the production of LC^3^ (CEM II/C-M (Q-LL) [56], NCC-3 and NCC-4 require sulfate adjustment to control the hydration properties of the blended LC^3^ system. 

The ternary blends of NCC, LP, and OPC achieved at least a “Level 1” reduction in greenhouse gas emissions.

## Figures and Tables

**Figure 1 materials-16-02684-f001:**
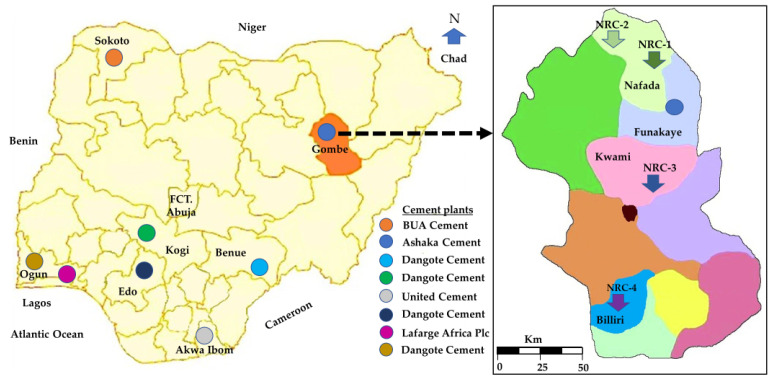
Map of Nigeria obtained from [42]. The location of different cement plants and of the sampling sites are added.

**Figure 2 materials-16-02684-f002:**
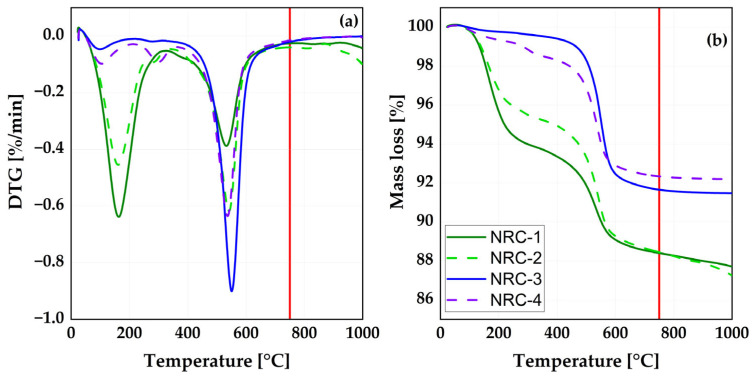
Differential thermal analysis (**a**) and mass loss (**b**) of the raw clays.

**Figure 3 materials-16-02684-f003:**
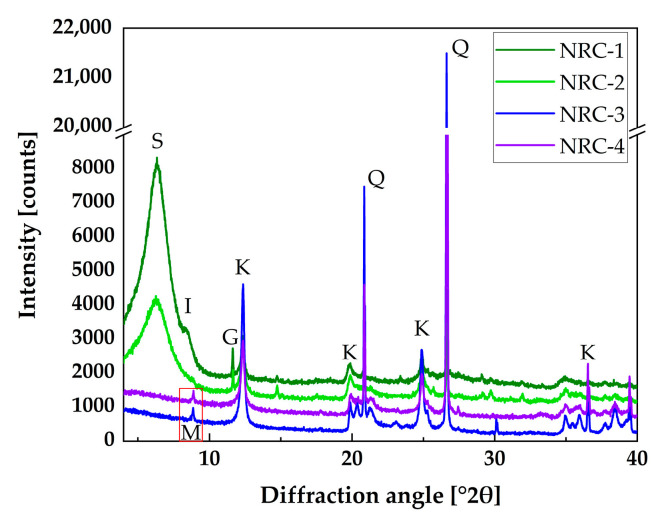
XRD patterns of the NRC: K—kaolinite, S—smectite, Q—quartz, M—mica, I—illite, and G—gypsum. The y-axes are shifted in the cases of NRC-1, 2, and 4 (the maximum shift for NRC-1 was 1000 counts).

**Figure 4 materials-16-02684-f004:**
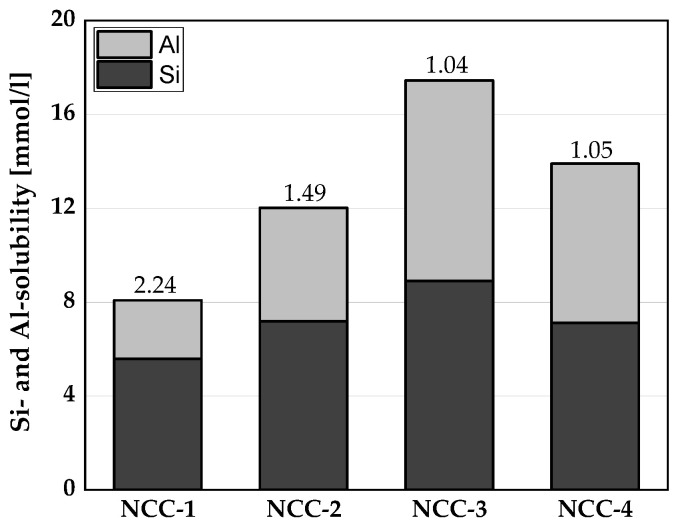
Si and Al ion solubilities of the CC.

**Figure 5 materials-16-02684-f005:**
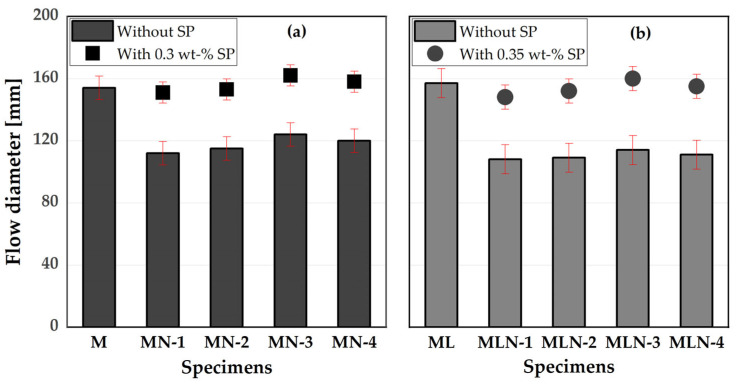
Influence of CC and LP on the flow spread of the mortar: (**a**) 20 vol.% CC + 80 vol.% OPC; (**b**) 30 vol.% CC + PLC.

**Figure 6 materials-16-02684-f006:**
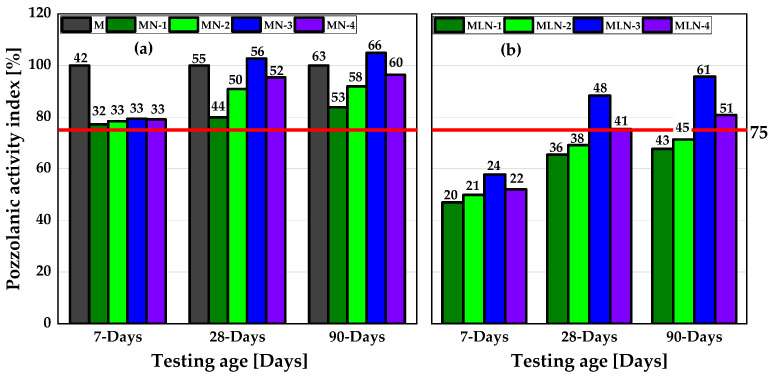
SAI of the mortars: (**a**) 20 vol.% CC + 80 vol.% OPC; (**b**) 30 vol.% CC + 15 vol.% LP and 55 vol.% OPC. The numbers placed above the bars denote the compressive strength value in MPa, approximated to the nearest whole number.

**Figure 7 materials-16-02684-f007:**
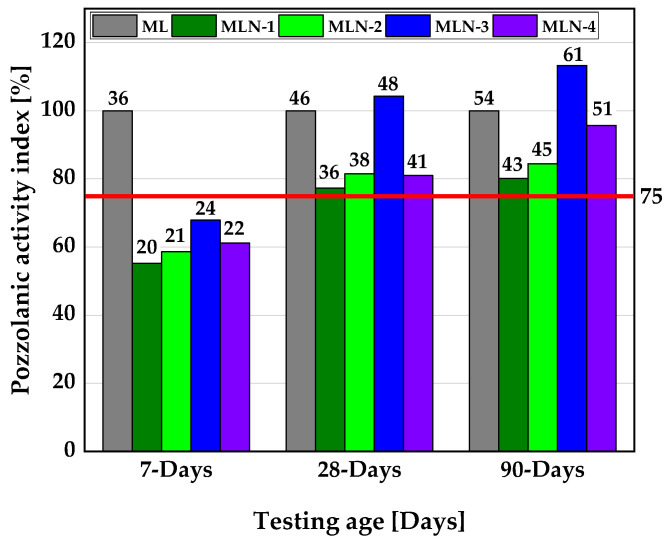
SAI of the NCC mortars compared to PLC mortar mixes. The numbers placed above the bars denote the compressive strength value in MPa, approximated to the nearest whole number.

**Figure 8 materials-16-02684-f008:**
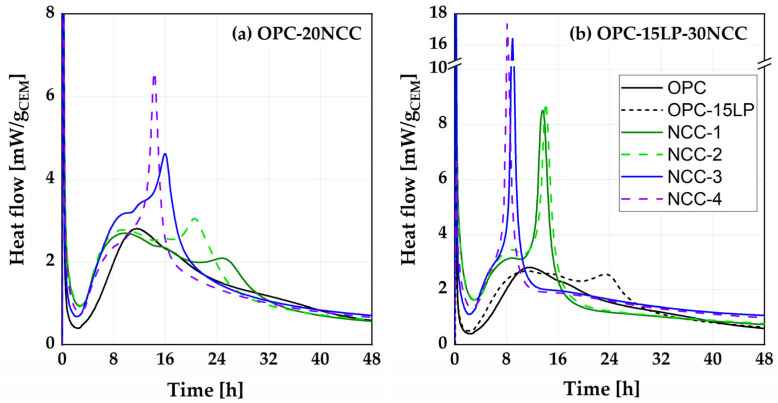
Heat flow of OPC-20NCC (**a**) and of OPC-15LP-30NCC (**b**) systems.

**Figure 9 materials-16-02684-f009:**
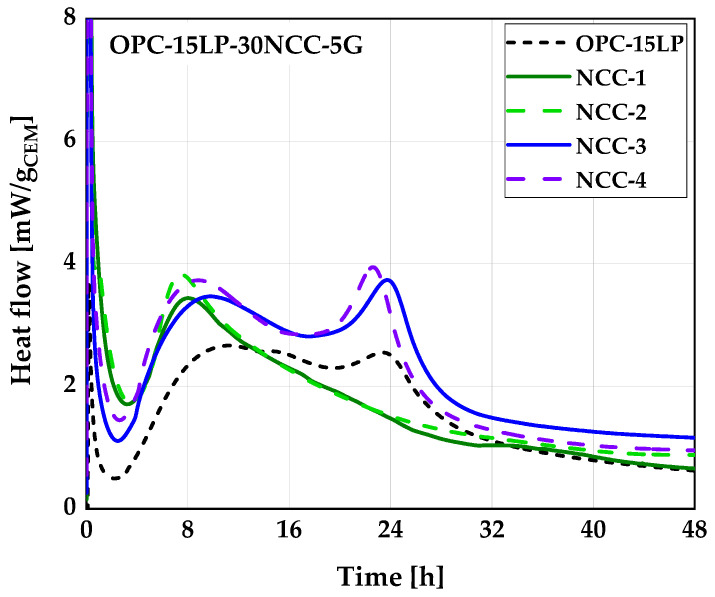
Heat flow of the OPC-15LP-30NCC systems with 5 wt.% G.

**Table 1 materials-16-02684-t001:** Mortar mix designation.

Mix Designation	OPC [g]	LP [g]	CC [g]	Sand [g]	Water [g]	SP [%]
M	450	-	-	1350	225	-
MN-1	360	-	76	1350	225	0.30
MN-2	360	-	80	1350	225	0.30
MN-3	360	-	79	1350	225	0.30
MN-4	360		81	1350	225	0.30
ML	383	57	-	1350	225	-
MLN-1	248	57	113	1350	225	0.35
MLN-2	248	57	121	1350	225	0.35
MLN-3	248	57	118	1350	225	0.35
MLN-4	248	57	121	1350	225	0.35

**Table 2 materials-16-02684-t002:** Mineralogical composition of the raw clays (wt.%).

Materials	Kaolinite	Smectite	Quartz	Mica/Illite	Rutile/Anatase	Hematite	Gypsum
NRC-1	13	71	6	10	-	-	<1
NRC-2	22	68	4	-	2	-	4
NRC-3	56	-	34	6	3	-	-
NRC-4	49	-	42	6	2	1	-

**Table 3 materials-16-02684-t003:** Oxide composition of the CC (wt.%).

Materials	SiO_2_	Al_2_O_3_	CaO	MgO	Fe_2_O_3_	TiO_2_	K_2_O	Na_2_O	SO_3_	LOI
NCC-1	55.0	19.1	1.3	3.1	7.7	1.0	1.8	8.6	0.8	1.6
NCC-2	54.3	24.5	1.9	2.1	9.5	1.1	1.5	0.2	1.9	3.0
NCC-3	68.1	24.7	0.1	0.1	2.5	3.5	0.4	<0.1	<0.1	0.5
NCC-4	68.2	20.4	0.2	0.2	8.2	1.5	0.7	<0.1	<0.1	0.6

**Table 4 materials-16-02684-t004:** Physical properties of the OPC, LP, and CC.

Materials	Specific Surface Area [m^2^/g]	Water Demand [wt.%]	Density [g/cm^3^]	d_10_ [µm]	d_50_ [µm]	d_90_ [µm]
OPC	1.0	28.9	3.17	2.6	16.0	42.8
LP	1.6	20	2.71	0.8	4.6	20.7
NCC-1	95.9	37.3	2.66	5.4	34.5	155.0
NCC-2	75.4	35.1	2.83	2.9	21.7	85.8
NCC-3	16.7	31.6	2.78	2.3	13.8	61.0
NCC-4	32.0	32.5	2.85	3.6	28.2	72.3

**Table 5 materials-16-02684-t005:** Hydration heat [J/g_CEM_] of OPC-20NCC and OPC-15LP-30NCC(+5G) systems after 48 h.

Specimens	OPC-20NCC	OPC-15LP-30NCC	OPC-15LP-30NCC-5G
OPC	254	-	-
PLC	-	270	-
NCC-1	300	349	346
NCC-2	307	332	349
NCC-3	320	361	426
NCC-4	293	340	426

## Data Availability

Not applicable.

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
