# Peer review of "Calcined Clays from Nigeria—Properties and Performance of Supplementary Cementitious Materials Suitable for Producing Level 1 Concrete"

_materials, 2023, doi:10.3390/ma16072684_

Round 1

Reviewer 1 Report

In this paper, the authors have characterized four samples of clays around Ashaka cement production plant, located in Gombe State, Nigeria.

The paper is poorly designed, and I have strong reservation on the results reported.

Paragraphs 2.2 and 2.3 report the description of the experimental results in the “Materials and methods” section) and it should be advisably moved in the “Result and discussion” section.

In the “Material and Methods” section, authors describe the use of X-ray diffraction for the identification of minerals and quantification of crystalline phases via Rietveld method. No diffraction patterns nor results are reported in the manuscript. The thermogravimetric analyses are briefly described.

The comprehensive discussion of data should help the reader to understand the impact of the present work and place these new discoveries in a broad contest. The discussion is limited and as the diffraction data are omitted the conclusions are not straightforward to support the hypothesis of this study.

The abbreviations should be defined at first mention. Most figure and table are not numbered in the text. Several references are missing.

Overall, I don’t see the potentiality of this study. For these reasons, I do not recommend the publication in the present form.

Author Response

Thank you very much for your comments on our manuscript. Please find below the answers to the comments. You will find all changes based on your advice in the revised document.

In this paper, the authors have characterized four samples of clays around Ashaka cement production plant, located in Gombe State, Nigeria.

The paper is poorly designed, and I have strong reservation on the results reported.

Author Response

Paragraphs 2.2 and 2.3 report the description of the experimental results in the “Materials and methods” section) and it should be advisably moved in the “Result and discussion” section.

TG and OPL and LP physical characterization were moved to the result section

In the “Material and Methods” section, authors describe the use of X-ray diffraction for the identification of minerals and quantification of crystalline phases via Rietveld method. No diffraction patterns nor results are reported in the manuscript. The thermogravimetric analyses are briefly described.

XRD patterns were added

The comprehensive discussion of data should help the reader to understand the impact of the present work and place these new discoveries in a broad contest. The discussion is limited and as the diffraction data are omitted the conclusions are not straightforward to support the hypothesis of this study.

The abbreviations should be defined at first mention. Most figure and table are not numbered in the text. Several references are missing.

This was checked and corrected

Overall, I don’t see the potentiality of this study. For these reasons, I do not recommend the publication in the present form.

Reviewer 2 Report

The utilization of supplementary cementitious materials continues to be a crucial issue and at the same time a focus of many researcher. I find this manuscript to provide a great insight how this can be done with the utilization of four naturally occurring clays. 

I recommend that this manuscript undergoes some changes in order to be improved and considered for publication.

- the authors need to express the novelty of this work in more explicit and concise manner (compared to other literatures).

- I suggest (for consideration) to remove the map of locations of different cement plants, and instead place XRPD results (besides table 3). 

- There is an error (see section 3.1.2. Reactivity assessment, section 3.2. Influence of NCC, see line 247, 282, ... please compare with references section. 

Author Response

Thank you very much for your comments on our manuscript. Please find below the answers to the comments. You will find all changes based on your advice in the revised document.

The utilization of supplementary cementitious materials continues to be a crucial issue and at the same time a focus of many researcher. I find this manuscript to provide a great insight how this can be done with the utilization of four naturally occurring clays. 

I recommend that this manuscript undergoes some changes in order to be improved and considered for publication.

Author Response

- the authors need to express the novelty of this work in more explicit and concise manner (compared to other literatures).

Research gap was added

- I suggest (for consideration) to remove the map of locations of different cement plants, and instead place XRPD results (besides table 3). 

XRD patterns were added

- There is an error (see section 3.1.2. Reactivity assessment, section 3.2. Influence of NCC, see line 247, 282, ... please compare with references section. 

This was checked and corrected

Reviewer 3 Report

Very interesting work. However, a few points have to be addressed:

1. A large number of omissions within the text;

- line 118 (K). Please, use Celsius (the units should be uniform within the text);

- problem with the reference numbers (lines 216, 225, 247, 282);

- line 267 - Figure . (which Figure?);

- line 313 (      displays);

- line 326 revealS;

- line 362 - Figure . (which Figure?);

- line 379 - helps TO reduce;

2. Lines 374 - 377. You state that the concrete grade is sufficient to be used for general construction works in sub-Saharan African region. Please, be more specific. Prove it with numbers, that it matches certain requirements for that region, make a comparison. Is that region seismic? Such a statement requires more prove. 

Author Response

Thank you very much for your comments on our manuscript. Please find below the answers to the comments. You will find all changes based on your advice in the revised document.

Very interesting work. However, a few points have to be addressed:

Author Response

  1. A large number of omissions within the text;

- line 118 (K). Please, use Celsius (the units should be uniform within the text);

We have changed K to °C to have uniform units like you suggested

- problem with the reference numbers (lines 216, 225, 247, 282);

This was checked and corrected

- line 267 - Figure . (which Figure?); the number of the figure was added

- line 313 (      displays);  figure was added

- line 326 revealS; corrected

- line 362 - Figure . (which Figure?); figure was added

- line 379 - helps TO reduce; corrected

  1. Lines 374 - 377. You state that the concrete grade is sufficient to be used for general construction works in sub-Saharan African region. Please, be more specific. Prove it with numbers, that it matches certain requirements for that region, make a comparison. Is that region seismic? Such a statement requires more prove. 

Round 2

Reviewer 1 Report

The Authors improve the manuscript materials-2271795 entitled “Calcined clays from Nigeria — Properties and performance of supplementary cementitious materials suitable for producing Level 1 concrete”. by addressing most of the comments. The manuscript needs to be organized in a better way to facilitate reading’s flow. The XRD data interpetation is not straightworward. I have further suggestions to improve the quality of the manuscript. For this reason, I suggest the acceptance of this manuscript with minor revisions.

Please consider the following comments:

Line 112. Remove Methods from “3. Materials and Methods”. You have a new section “4. Methods”.

Line 127- 148. Move paragraphs “Calcination and grinding of NRC” and “Characterization of the cement, limestone powder and superplasticizer” in section “4. Methods”.

Line 136-138. Figure 2 is out of place. Why it is in the methods, and it is not numbered? It should move in “Results and discussion” and described in the text.

Line 153-154. Move this sentence in “Results and discussion”.

Line 160. Change “employed” with “performed.”

Line 160. How do you know that there is no significant amount of amorphous? You did not calculate it with Rietveld method.

Figure 3 (line 236). How did you resolve the large peak of smectite? Smectite is a family of swelling clays. Do you have vermiculite or montmorillonite? Also, given the broad peak, you can have a mixed-layered clay. The shoulder at the right side of the smectite peak that you labelled M = mica is more likely illite. It seems that you have to distinguish the smectite-illite components before to perform the Rietveld method. Anatase is an obsolete term, please change it with rutile.

Table 2 (line 240). Remove mica and leave illite. Remove anatase and leave rutile.

Author Response

Dear reviewer,

thanks again for your second and sorrow review of our manuscript. Please find below the answers to the comments.

Reviewer 1 Report

The Authors improve the manuscript materials-2271795 entitled “Calcined clays from Nigeria — Properties and performance of supplementary cementitious materials suitable for producing Level 1 concrete”. by addressing most of the comments. The manuscript needs to be organized in a better way to facilitate reading’s flow. The XRD data interpretation is not straightforward. I have further suggestions to improve the quality of the manuscript. For this reason, I suggest the acceptance of this manuscript with minor revisions.

Please consider the following comments:

Author Response

Line 112. Remove Methods from “3. Materials and Methods”. You have a new section “4. Methods”.

“Methods” was removed

Line 127- 148. Move paragraphs “Calcination and grinding of NRC” and “Characterization of the cement, limestone powder and superplasticizer” in section “4. Methods”.

The paragraphs were moved

Line 136-138. Figure 2 is out of place. Why it is in the methods, and it is not numbered? It should move in “Results and discussion” and described in the text.

Figure 2 was provided and described in detail under 4.1.1

Line 153-154. Move this sentence in “Results and discussion”. Done

Line 160. Change “employed” with “performed.” Changed

Line 160. How do you know that there is no significant amount of amorphous? You did not calculate it with Rietveld method.

The calculations with an external standard (Zircon, G-factor method) showed a content of crystalline phases of about 100%. For this reason the authors decided to use the normalized Rietveld quantification in the result section.

Figure 3 (line 236). How did you resolve the large peak of smectite? Smectite is a family of swelling clays. Do you have vermiculite or montmorillonite? Also, given the broad peak, you can have a mixed-layered clay. The shoulder at the right side of the smectite peak that you labelled M = mica is more likely illite. It seems that you have to distinguish the smectite-illite components before to perform the Rietveld method.

You are right. We have a montmorillonite. It wasn´t our intention to label the shoulder of the smectite peak with “mica”. The “M” just belongs to the two kaolinite-rich clays NRC3 and NRC4. We encased the peak in question to make it clear directly inside the graph and labeled the shoulder with “I” for Illite. As the quantification of NRC1 shows, we distinguish between smectite and illite.

Thank you for your advice to make the results more clearly.

Anatase is an obsolete term, please change it with rutile.

We have both modifications of TiO2 in our samples. Anatase and rutile are not the same.

Table 2 (line 240). Remove mica and leave illite. Remove anatase and leave rutile.

Sorry, but we cannot agree with these suggestions. As the figure above confirms that we have both rutile and anatase. The situation is similar for mica and illite. In case of the kaolinite-rich clays we have mica in our samples and in case of NRC1 (shoulder at the smectite peak) we have an illite.
